# Nanobubbles explain the large slip observed on lubricant-infused surfaces

Christopher Vega-Sánchez[1,2,3], Sam Peppou-Chapman[1,2], Liwen Zhu[1,2] & Chiara Neto [1,2 ✉]

Lubricant-infused surfaces hold promise to reduce the huge frictional drag that slows down the flow of fluids at microscales. We show that infused Teflon wrinkled surfaces induce an effective slip length 50 times larger than expected based on the presence of the lubricant alone. This effect is particularly striking as it occurs even when the infused lubricant's viscosity is several times higher than that of the flowing liquid. Crucially, the slip length increases with increasing air content in the water but is much higher than expected even in degassed and plain Milli-Q water. Imaging directly the immersed interface using a mapping technique based on atomic force microscopy meniscus force measurements reveals that the mechanism responsible for this huge slip is the nucleation of surface nanobubbles. Using a numerical model and the height and distribution of these surface nanobubbles, we can quantitatively explain the large fluid slip observed in these surfaces.

[1] School of Chemistry, The University of Sydney, Sydney, NSW 2006, Australia. [2] University of Sydney Nano Institute, The University of Sydney, Sydney, NSW 2006, Australia. [3] School of Electromechanical Engineering, Costa Rica Institute of Technology, Campus Cartago, Cartago 159-7050, Costa Rica. ✉email: chiara.neto@sydney.edu.au

When a liquid flows through a channel, its velocity at the channel wall is reduced as a consequence of its interaction with the wall, an effect called frictional drag. In macroscopic flows, a no-slip boundary condition is usually assumed, i.e., the liquid relative velocity is expected to be zero at the wall[1]. However, in the past two decades evidence of nanoscale interfacial slip has emerged in situations when the flow is highly confined and the wettability of the solid by the liquid is poor[2,3]. Slip is quantified using the slip length $b$, the distance beyond the interface at which the liquid velocity linearly extrapolates to zero. The larger the slip length $b$, the larger the reduction in frictional drag.

Superhydrophobic[4,5] and lubricant-infused surfaces (LIS)[6–8] have been shown to reduce drag substantially[9–15], which makes them attractive to reduce the energy required to drive flow. Drag reduction by these surfaces is explained with the reduced contact area between the solid and the fluid[9], which results in an "apparent slip" of the flowing liquid of viscosity $\mu_w$ over the air (in the case of superhydrophobic surfaces) or over a lubricant of lower viscosity $\mu_o$ in LIS, compared to the case of a solid surface. For a continuous lubricant film of thickness $h_o$, the maximum apparent slip length, defined from the fluid-lubricant interface, should depend on viscosity ratio $\mu_w/\mu_o$[16,17]:

$$b_{\max} = \frac{\mu_w}{\mu_o} h_o \qquad (1)$$

The gas layer (plastron) present on immersed superhydrophobic surfaces naturally produces large (larger than several hundred nm) slip length values in water, as the viscosity ratio is around 55[18]. However, recent studies have reported similarly large slip length values also in LIS, with values that are 3 times[10], 7 times[11], and 90 times[12] higher than predicted by the apparent slip model (Eq. (1)). This anomalous slip is especially hard to explain in situations where $\mu_w/\mu_o \leq 1$, i.e., when the lubricant viscosity is higher than (or close to) that of the flowing liquid. No explanation has been provided for this discrepancy, other than the low sensitivity of experimental measurements conducted at a low viscosity ratio[10] or the large variation in the surface roughness resulting in large uncertainty in the microchannel height estimation needed to quantify the slip length[11]. The aim of this work is to investigate the mechanism leading to the unexpectedly large interfacial slip on lubricant-infused surfaces.

Here, we report accurate measurements of pressure drop along a microfluidic channel incorporating lubricant-infused surfaces to estimate the effective slip length of these surfaces (Fig. 1a). The flow of water ($\mu_w = 0.89$ mPa s) was investigated over nanowrinkled Teflon surfaces (Fig. 1b, c), both in a superhydrophobic state (no lubricant) and infused with silicone oil 10 cSt ($\mu_o = 9.30$ mPa s). The average lubricant thickness within the surface roughness, measured using atomic force microscopy (AFM) meniscus force mapping[19], was found to be $h_o \sim 1$ μm. For the flow of Milli-Q water over LIS, the effective slip length was $3.8 \pm 1.9$ μm, a value that is 38 times higher than predicted by the apparent slip model and in agreement with previous observations[10–12]. For this slip length value, the apparent slip model would predict a lubricant film of $h_o \sim 38$ μm, which is clearly incorrect given the channel is only 100 μm tall. The meticulous minimization of all sources of error[20] and extensive experimental validation enables us to highlight this discrepancy with confidence (further details are contained in "Methods" and in the Supporting Information).

Underwater mapping of the infused surfaces revealed the nucleation of nanobubbles of the thickness of the order of 100 nm in plain Milli-Q water ($c_{air} \sim 23.0 \pm 0.3$ mg kg$^{-1}$), and the magnitude of measured slip was seen to increase with increasing air content $c_{air}$ in the flowing liquid (Fig. 1d, e). When flowing gassed water ($c_{air} \sim 44 \pm 4$ mg kg$^{-1}$), the effective slip was estimated to be $5.2 \pm 1.1$ μm. At air content higher than this, the size of the bubbles reached the microscale, making the pressure drop measurements impossible. By comparison, no-slip was measured when testing either smooth hydrophilic (silicon wafer) or smooth hydrophobic surfaces (Teflon; octadecyltrichlorosilane OTS-coated silicon wafer) and smooth infused surfaces (grafted-PDMS on a silicon wafer) under the same conditions (Fig. 1e).

Nanobubble nucleation in LIS has not been considered before, as: (1) the assumption is that when the lubricant is depleted, water (the flowing liquid) immediately fills the gaps; (2) acquiring experimental evidence of nanobubbles (particularly on a structured surface) is complex. Based on our results, the gas nucleation process is not unique to our infused wrinkled Teflon surfaces. It is expected to occur in all LIS with similar characteristics (rough and hydrophobic surfaces).

In the remainder of the article, our aim is to systematically investigate the mechanism of interfacial slip on smooth and wrinkled lubricant-infused surfaces and provide evidence of gas nucleation in these surfaces. Our ability to map at the same time the lubricant film and the nanobubbles underwater enabled us to identify the unexpected nucleation of nano- and microbubbles in lubricant-infused surfaces. The bubbles locally displace the lubricant and are pinned to the underlying Teflon wrinkles. Using a numerical model and the experimentally mapped distribution and height of nanobubbles, we show that our measurements of slip are in good agreement with the fluid mechanic's theory only if the gas bubbles are included. Our findings provide a quantitative explanation for the large slip reported in LIS in our and previous works[10–12].

## Results

**Surface properties of the tested substrates**. Using the microfluidic setup shown in Fig. 1a, the flow of water and water–glycerol solutions with different content of air $c_{air}$ was studied. The air content was adjusted by placing the working fluid under an atmosphere of air at specific pressures. Its concentration was estimated by direct measurement of the dissolved oxygen concentration (see "Methods" and Supporting Information). A number of smooth and nanowrinkled surfaces (Table 1) were tested as bottom surfaces in the channels: hydrophilic silicon wafer, smooth hydrophobized silicon wafer (OTS-coated wafer with RMS roughness of 1.6 nm) both as prepared and infused with silicone oil, grafted silicone on silicon wafer[21] (PDMS-Si wafer with RMS roughness of 0.6 nm), plain Teflon sheet, superhydrophobic Teflon wrinkles[22], and Teflon wrinkles infused with silicone oil or hexadecane[23].

Water contact angle and sliding angle values were used as qualitative measures of LIS properties and successful infusion. Silicone oil-infused Teflon wrinkles exhibit the lowest contact angle hysteresis and sliding angle (both $3 \pm 1°$) of all the tested surfaces, which is an indication of stable lubricant infusion. The relatively high contact angle hysteresis on the superhydrophobic Teflon wrinkles suggests a partially collapsed Wenzel state.

As shown in Fig. 1b, c, the Teflon wrinkles exhibit a hierarchical topography uniform over centimeter-scale areas: small nanowrinkles on top of larger-scale features. The nanowrinkles have a width of $180 \pm 40$ nm and a height of $220 \pm 50$ nm. The larger features have an average peak-to-valley height of $\sim 790 \pm 210$ nm with a bimodal characteristic length $s$ (Fig. 1d) of 2.5 and 13 μm (Supplementary Fig. S5). The wrinkles are superhydrophobic as prepared and, in order to make them LIS, they are infused with one of three lubricants prior to being placed in the microfluidic device: silicone oil (10 cSt and 5 cSt) and hexadecane (3.88 cSt). Hydrophobized silicon wafers were infused only with silicone oil 10 cSt.

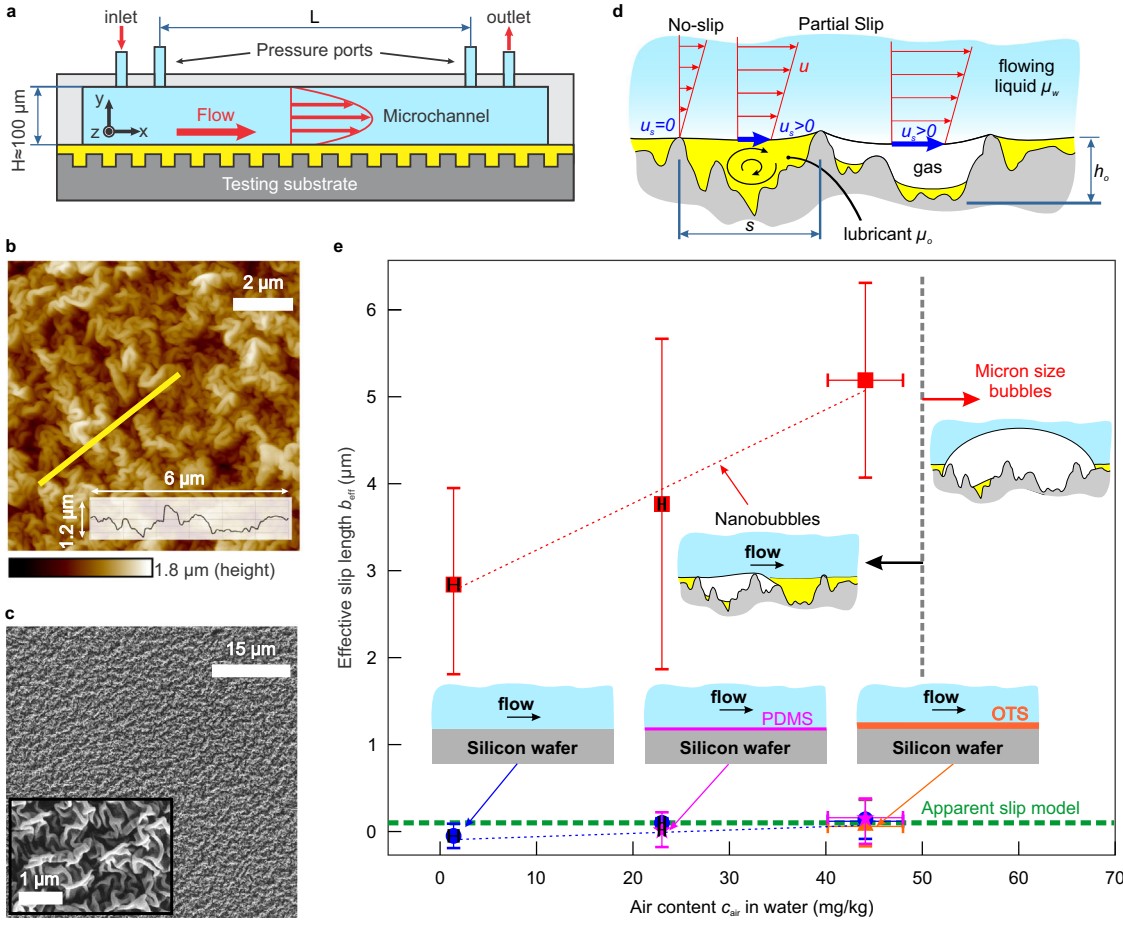

**Fig. 1 Estimation of the effective slip length of water flowing over different substrates using a microfluidic channel. a** Schematic diagram of the microfluidic flow cell used in this study (not to scale). Water (blue) flows from the inlet to the outlet over infused lubricant (yellow). **b, c** Atomic force and scanning electron micrographs of the employed Teflon-wrinkled surfaces. **d** Schematic of flow over a lubricant-infused wrinkled surface containing a nanobubble, with definition of variables. **e** Effective slip length measured for water flowing on Teflon-wrinkled surfaces infused with silicone oil 10 cSt (red squares), on smooth silicon wafer (blue circles), hydrophobized (OTS-coated) silicon wafer (orange triangle), and grafted-PDMS on silicon wafer (purple stars), as a function of the air content $c_{air}$ in the water (further details are contained in "Methods" and in the Supporting Information). Vertical error bars are standard deviation over at least 7 (and up to 22) repetitions using different microfluidic channels and different microfluidic devices (Supplementary Table S2). The horizontal error bars smaller than the size of the symbols are shown in black. The statistical difference between the data points for the Teflon-wrinkled surfaces (red squares) is significant (the comparison between each pair of data points produces $P$ values of $4 \times 10^{-2}$, $10 \times 10^{-5}$ and $4 \times 10^{-13}$). The slip predicted from Eq. (1) is shown (green dashed line) for $h_o = 1 \, \mu m$. Source data are provided as a Source Data file.

**Table 1 Advancing (ACA) and receding contact angle (RCA), sliding angle (SA), and contact angle hysteresis (CAH) values for 10 μL Milli-Q water droplets in air on all the tested substrates.**

| Substrate | ACA (°) | RCA (°) | SA (°) | CAH (°) |
|---|---|---|---|---|
| Silicon wafer (control) | Pinning | Pinning | >90 | – |
| OTS-silicon wafer | 115 ± 2 | 102 ± 1 | 29 ± 1 | 13 ± 1 |
| OTS-silicon wafer infused | 110 ± 2 | 107 ± 2 | 5 ± 2 | 3 ± 2 |
| PDMS-silicon wafer | 112 ± 2 | 100 ± 2 | 9 ± 2 | 12 ± 2 |
| Plain Teflon sheet | 121 ± 9 | 96 ± 7 | 29 ± 2 | 25 ± 7 |
| TW (Teflon wrinkles) | 169 ± 5 | 154 ± 3 | 10 ± 2 | 15 ± 4 |
| TW infused (silicone oil) | 113 ± 3 | 110 ± 3 | 3 ± 1 | 3 ± 1 |
| TW infused (hexadecane) | 109 ± 3 | 98 ± 2 | 10 ± 2 | 11 ± 2 |

Mean value and standard deviation are shown.

**Nanoscale mapping of the lubricant layer.** Equation (1) defines the upper bound for the slip length expected for a continuous lubricant film of thickness $h_o$. To enable a correct theoretical prediction, the value of $h_o$ needs to be known accurately, which is challenging on structured surfaces. Most authors in the literature assume that the lubricant layer thickness is equal to the height of the surface roughness[8]. Still, this approach is unrealistic because it presumes a flat interface pinned at the top of the structures, ignoring the deformation of the fluid-lubricant interface caused by interfacial tension and long-range forces, and the potential lubricant depletion due to the shear stress imposed by the external fluid.

Here, the lubricant film thickness $h_o$ was mapped with nanoscale resolution using AFM meniscus force measurements[19]. Simultaneously mapping the wrinkled surface topography (Fig. 2a) and the oil thickness (Fig. 2b) provides a complete description of the oil distribution within the surface topography and allows to quantify the volume of lubricant retained by the substrate between experiments. Before exposure to flow, the average lubricant film thickness $h_o$ was of the order of 1 μm, corresponding to a total volume of lubricant in the microfluidic channel of ~100 nL. After shearing at 600 μL min⁻¹ for 30 min, the thickness of the remaining lubricant was 2–5 nm on the wrinkle tops (dark regions in Fig. 2b) and larger than 300 nm in the valleys (white regions in Fig. 2b), corresponding to an average

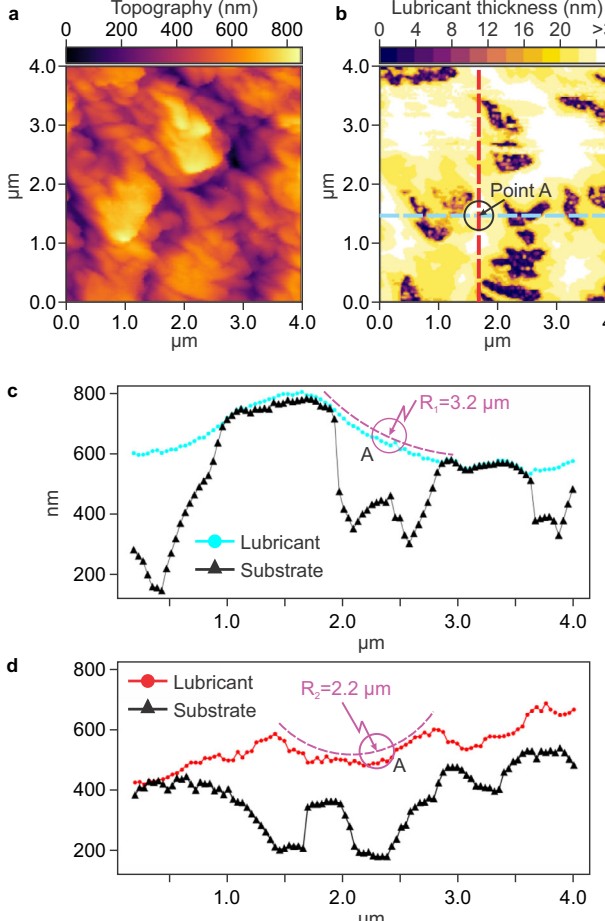

**Fig. 2 Characterization of Teflon wrinkles infused with silicone oil (10 cSt) using meniscus force atomic force microscopy. a** Topographical map of an infused Teflon-nanowrinkled surface in air. **b** Silicone oil thickness distribution on the surface after shearing at a flow rate of 600 μL min$^{-1}$ for 30 min. The color scale is not linear: darker color represents regions with a lubricant film less than 10-nm thick, white regions a lubricant thickness of more than 300 nm. **c** Profile of the cross-section corresponding to the horizontal blue line in panel (**b**), and (**d**) profile of the vertical red line in panel (**b**). $R_1$ and $R_2$ are the principal radii of curvature at point A. Source data are provided as a Source Data file.

$h_o = 330 \pm 230$ nm. The solid fraction $\phi_s$ exposed (regions with lubricant thickness lower than 10 nm) on the Teflon wrinkles was less than 2% at 200 μL min$^{-1}$, and increased to 16% at 600 μL min$^{-1}$ (Supplementary Fig. S8).

The AFM mapping of the lubricant interface (Fig. 2c, d) enables estimating the local capillary pressure responsible for the oil retention within the surface; from the curvature at point A ($7.7 \times 10^5$ m$^{-1}$), the Laplace pressure was estimated to be of the order of $10^4$ Pa, which is about two orders of magnitude higher than the applied static pressure in our experiments, confirming the strength of the lubricant trapping.

**Measurements of slip length using microchannels.** Using two different rigid microfluidic devices at monitored temperature (Supplementary Fig. S1), highly accurate measurements of the pressure drop $\Delta p$ across microchannels were made as a function of flow rate from 200 to 800 μL min$^{-1}$ (pressure drop versus flow rate method[1,24]). The experimental uncertainty in $\Delta p$ was minimized to ~ 2% by reducing the error associated to individual measurements of flow rate, static pressure, temperature, and

channel height (further details are contained in "Methods" and in the Supporting Information). The microchannels have height $H \approx 105$ μm (measured individually after each experiment, Supplementary Fig. S3), width $W = 2.5$ mm, and length $L = 25$ mm. At each flow rate, if slip occurs on the sample surface, the measured pressure drop $\Delta p$ is lower than that expected under no-slip conditions $\Delta p_{\text{no−slip}}$, which is calculated from the standard friction factor equation[25]:

$$\Delta p_{\text{no−slip}} = \frac{f}{\text{Re}} \frac{\rho U^2 L}{2 D_h}, \tag{2}$$

where Re is the Reynolds number, $\rho$ is the fluid density, $U$ is the average velocity of the fluid, $L$ is the channel length, $D_h$ is the hydraulic diameter of the channel, and $f$ is the friction factor[25]. By comparison of the experimental $\Delta p$ and $\Delta p_{\text{no−slip}}$, the drag reduction factor is estimated from:

$$D_R = \frac{\Delta p_{\text{no−slip}} - \Delta p}{\Delta p_{\text{no−slip}}} \tag{3}$$

An effective slip length $b_{\text{eff}}$, which represents an average of the slip over the whole channel, is computed as[11]:

$$b_{\text{eff}} = \frac{H D_R}{3 - 4 D_R} \tag{4}$$

The microfluidic setup was validated by flowing water on smooth plasma-cleaned silicon and OTS-coated silicon wafers (OTS-Si). On these substrates, no drag reduction was expected nor observed (Fig. 1e and Supplementary Fig. S4), as the slip length is too small to be measured with a microfluidic setup (around 24 nm for both silicone oil-infused OTS-Si[14] and non-infused OTS-Si[26], as measured by colloid probe AFM). Similar results were obtained for the PDMS-Si, as shown in Fig. 1e. Although the OTS and grafted-PDMS surfaces are hydrophobic with low contact angle hysteresis, they are not capable of reducing drag beyond the nanoscale[14].

**Effective slip of water as working fluid.** A significant reduction in the pressure drop was measured upon flowing water over infused Teflon wrinkles ($0.1 \leq \mu_w/\mu_o \leq 0.3$). The obtained drag reduction increased with increasing air content in the water, as shown in Fig. 3. For the highest air content tested ($c_{\text{air}} \sim 44 \pm 4$ mg kg$^{-1}$), the average slip length was found to be $5 \pm 1$ μm corresponding to a drag reduction of $12 \pm 3$%, while for lowest air content water ($c_{\text{air}} \sim 1.4 \pm 0.5$ mg kg$^{-1}$) the slip length was, on average, $3 \pm 1$ μm corresponding to an 8% of reduction of drag. For each air content value $c_{\text{air}}$, the effective slip values do not vary significantly under the tested flow rates (Fig. 3c), similar to previous observations[11,12]. Figure 3b shows the pressure drop reduction as a function of the flow rate for different substrates, using water with the highest air content tested (44 mg kg$^{-1}$). Plain Teflon sheets showed no reduction in drag, while an average drag reduction factor of $15 \pm 3$% was observed for superhydrophobic Teflon wrinkles, $12 \pm 2$% for both hexadecane-infused and 10 cSt silicone oil-infused Teflon wrinkles. This converts into effective slip length values of $7 \pm 2$ μm for superhydrophobic Teflon wrinkles and $5 \pm 1$ μm for silicone oil-infused Teflon wrinkles. These values are at least five times larger than the surface roughness and, therefore, cannot be attributed to an error in the definition of the slip plane. Similar slip length values were obtained when the lubricant was silicone oil 5 cSt (blue circle in Fig. 3e) and hexadecane (3.88 cSt, black hexagons).

**Effective slip of water–glycerol mixtures as working fluids.** Glycerol–water mixtures with $c_{\text{air}} \sim 44 \pm 4$ mg kg$^{-1}$ were used as working fluids to increase the viscosity ratio ($\mu_w/\mu_o \geq 1$, Fig. 3c).

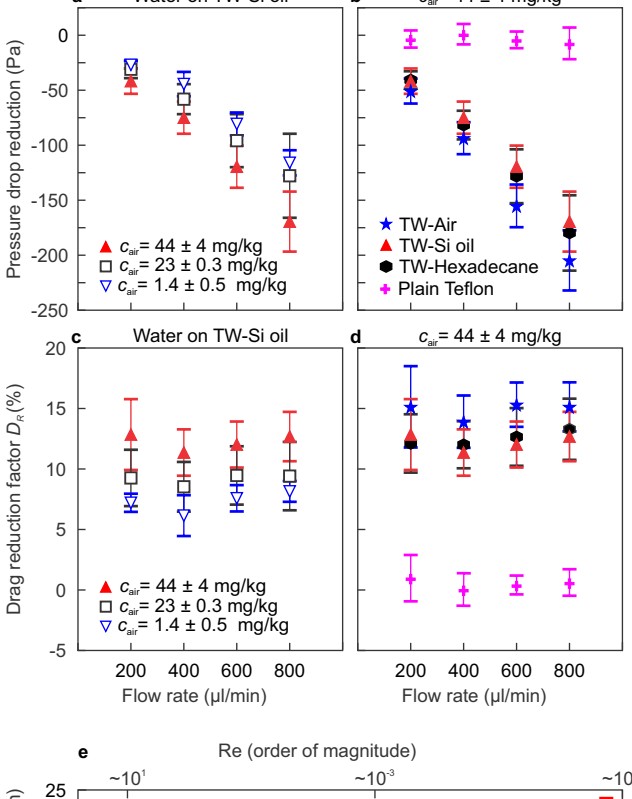

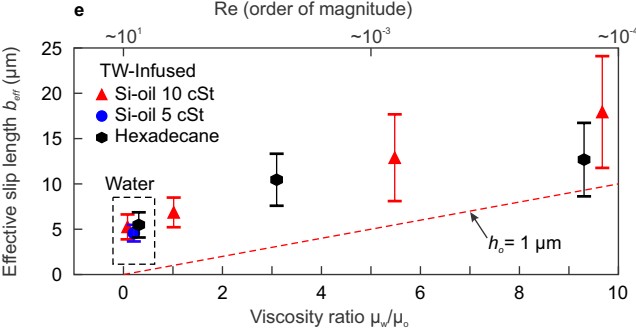

**Fig. 3 Microfluidic measurements and estimation of the drag reduction and effective slip length. a** Average pressure drop reduction measured for water over Teflon-wrinkled surfaces infused with silicone oil 10 cSt (TW-Si oil) vs flow rate, for water with different air content $c_{air}$. **b** Average pressure drop reduction vs flow rate, for water with air content $c_{air} \sim 44 \pm 4$ mg kg$^{-1}$ for superhydrophobic Teflon wrinkles (TW-air), TW infused with silicone oil 10 cSt and with hexadecane, and non-infused plain Teflon sheets. **c, d** Drag reduction factor vs flow rate of water over the same surfaces reported in parts (**a**) and (**b**), respectively. The symbols in (**d**) are the same as in (**b**). **e** Effective slip length vs viscosity ratio at constant $Ca = 0.001$ for Teflon wrinkles infused with silicone oil 10 cSt (red triangles) and 5 cSt (blue circle), and hexadecane (black hexagons). Water and glycerol–water mixtures with air content $c_{air} \sim 44 \pm 4$ mg kg$^{-1}$ were used as working fluid to achieve the shown viscosity ratio. The slip predicted from Eq. (1) is shown (red dashed line) for $h_o = 1$ µm. Each data point and error bar correspond to the average and standard deviation of at least 7 (and up to 22) repetitions using different microfluidic channels and different microfluidic devices (Supplementary Table S2). Source data are provided as a Source Data file.

In all the experiments, the capillary number was fixed at $Ca = \mu_w U/\gamma_{w/o} = 0.001$ to keep the same balance of viscous to interfacial stress. As expected from the apparent slip model, a further increase in the effective slip length was observed by increasing the viscosity ratio. The maximum drag reduction measured was $28 \pm 10\%$, obtained at viscosity ratio $\mu_w/\mu_o = 9.7$, corresponding to an effective slip length of $b_{eff} = 18 \pm 6$ µm. Both

silicone oil and hexadecane produced similar drag reduction when infused into the Teflon wrinkles.

The measured values of drag reduction remained constant over 24 h of flow (Supplementary Fig. S6). The lubricant thickness in the LIS only decreased stepwise when the flow rate was increased and then remained constant when the system was sheared for periods of 55 minutes (Supplementary Fig. S7). This excellent lubricant retention is due to the complete spreading of silicone oil on the Teflon wrinkles underwater[23,27], and the high negative Laplace pressure holding the lubricant in place (Fig. 2).

**Failure of apparent slip model to explain observed slip.** The measured slip length for the superhydrophobic Teflon wrinkles agrees well with the model presented by Ybert et al.[17] for surfaces containing posts: $\frac{b}{s} \simeq \frac{0.325}{\sqrt{\phi_s}} - 0.44$. For our superhydrophobic Teflon wrinkles, the model predicts slip lengths of $1 < b_{model} < 24$ µm for $2.5 < s < 13$ µm and $0.02 < \phi_s < 0.16$, and our measurement is $b_{eff} = 7 \pm 2$ µm. On the contrary, for our LIS the experimental $b_{eff}$ is 55, 7, and 2 times higher than that predicted by the apparent slip model for viscosity ratio $\mu_w/\mu_o$ of 0.1, 1, and 5.5, respectively. The theoretical slip length values are shown as a dashed line in Figs. 1e and 3e, calculated using lubricant thickness $h_o \approx 1$ µm. Given that in our experiments the lubricant layer is not of constant $h_o$ but rather of varying thickness and with $\phi_s \sim 2\%$ or higher, the measured slip length should be much lower than predicted by Eq. (1).

Based on the low uncertainty in the experimental results and their high reproducibility with different lubricants, flowing liquids and microfluidic cells, our results indisputably show that the large discrepancy between measurement and the apparent slip model at a low viscosity ratio is a real effect. Adjusting the lubricant thickness to give the best fit of the apparent slip model to the experimental data, the lubricant thickness obtained is unrealistic: for a viscosity ratio $\mu_w/\mu_o = 0.1$, the model predicts a lubricant thickness of $h_o = 50$ µm, half the height of the channel, which is clearly incorrect. With this same thickness, the model would predict a slip length $b \sim 500$ µm for viscosity ratio $\mu_w/\mu_o = 9.7$. Therefore, even though it is idealized for a uniformly thick film, the apparent slip model underestimates the effective slip length measured at the viscosity ratios tested.

**Mechanisms for slip on lubricant-infused surfaces.** The large slip observed in our experiments could be explained as a combination of three mechanisms: (i) molecular slip at the water–lubricant interface; (ii) recirculation within the lubricant pools; (iii) a larger effect due to the presence of a lubricant of much lower viscosity than the ones used here.

(i) This effect is expected to be nanoscale, as suggested by molecular dynamics simulations[28–30] and experiments[14,15], and not quantifiable with our experimental setup.

(ii) As the velocity and shear stress in the flowing fluid are expected to match those in the lubricant at the interface, the lubricant is forced to recirculate within the pools between the wrinkles, as shown in Fig. 1d. This is the standard approach to explain slip in LIS, in which the lubricant interfacial velocity manifests as an apparent slippage. Fluids flowing through spaces larger than a few tens of nanometers can be treated as a continuum medium[31]. Therefore, we assume our oil film of 1-µm thickness can be treated as a continuum. A two-dimensional numerical model was developed to study a Couette flow over a surface made of a profile of Teflon wrinkles extracted from AFM maps (Supplementary Fig. S15). The model was implemented using the commercial finite element software Comsol Multiphysics to solve the Navier–Stokes equations for both the water and the lubricant. Recirculation zones were indeed observed in

the pools between the Teflon wrinkles. However, as expected, the magnitude of the velocity derived from the numerical model (50 μm s$^{-1}$, see inset in Supplementary Fig. S15) is two orders of magnitude smaller than the interfacial velocity required to explain our experimental data. Furthermore, the recirculation of lubricant alone cannot achieve the same level of slip provided by a continuous lubricant film of the same thickness (the apparent slip model).

(iii) Finally, the presence of a lubricant of much lower viscosity, such as a thin air layer at the interface, as seen in superhydrophobic surfaces, would produce a significant reduction in the pressure drop, easily observable in our measurements[24]. As shown in "Methods" and in the Supporting Information, an air layer of the average thickness of 100 nm would be sufficient to explain our high slip results with water (at viscosity ratio $\mu_w/\mu_o = 0.1$ and air content of $c_{air} \sim 44 \pm 4$ mg kg$^{-1}$), and 10 nm air layers would explain the higher viscosity ratio results, as presented in Supplementary Fig. S14 and Supplementary Table S7. A continuous layer of air at the interface would be destabilized by intermolecular forces and surface tension[32]. Still, air bubbles could be stabilized by contact line pinning on exposed regions of the Teflon surfaces. As air preferentially wets the Teflon surface rather than water, nanobubbles can form from the air dissolved in the working fluid and in the lubricant and can be easily overlooked in microfluidic experiments.

**Evidence of nanobubble nucleation on the LIS**. The presence of nano- and microbubbles on the infused surfaces was demonstrated using underwater AFM mapping[33] and confocal microscopy. Silicone oil-infused wrinkled surfaces were immersed in Milli-Q water ($c_{air} \sim 23.0 \pm 0.3$ mg kg$^{-1}$) in an AFM liquid cell[27], where nanobubble nucleation was directly observed using AFM force spectroscopy and mapping over time (Fig. 4 and Supplementary Figs. S10–13). Figure 4a shows cantilever deflection-piezo displacement curves on Teflon wrinkles under Milli-Q water. When the AFM tip is distant from the substrate, the deflection is zero. On contact with the solid Teflon, the deflection increases steeply (Point 1). When the tip first contacts a lubricant layer (Point 2), an abrupt attraction is revealed by the negative deflection due to capillary attraction of the meniscus as the lubricant wets the AFM tip[19]. The separation between the hard contact and the jump-in points gives the thickness of the lubricant at this location. When the tip first contacts a bubble (Point 3), a positive slope is seen in the deflection-piezo displacement curve. This positive slope is the signature evidence of nanobubbles, consistent with extensive prior work[34–36]. Finally, when a bubble is present over a lubricant layer, both features can be distinguished in the force curve (Point 4). These force curves are representative of thousands of force curves obtained through the force mapping of the immersed surfaces (Supplementary Fig. S10; further details are contained in "Methods" and in the Supporting Information). The force curves were analyzed automatically using a Python script, which identified the nature of the lubricant/water and air/water interface.

In Fig. 4 (extracted from experimental measurements shown in Supplementary Fig. S10), the evolution of the gas nucleation process was mapped over time, as we continuously mapped the same region underwater, without repositioning the tip until the nanobubble appeared. In the lower-left corner of Fig. 4b (right panel), a nanobubble of average thickness 90 ± 70 nm and width around 2.5 μm can be seen over the infused substrate. The same signature positive deflection is observed in immersed superhydrophobic wrinkles (Supplementary Fig. S12). The profile of

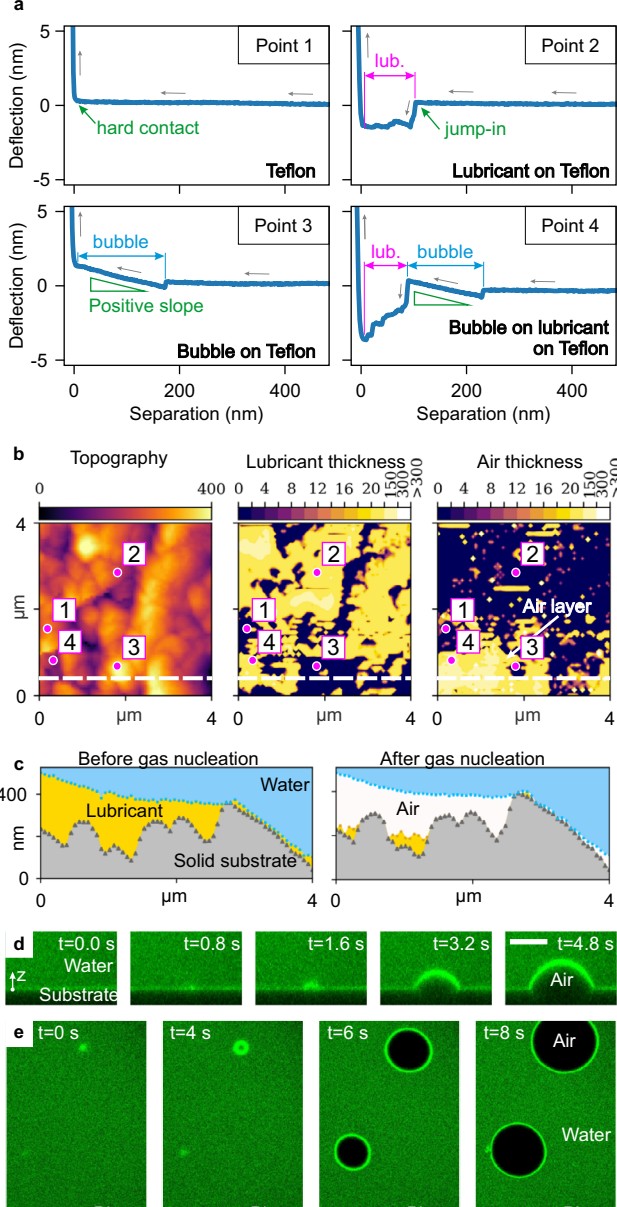

**Fig. 4 Evidence of nanobubble nucleation on lubricant-infused surfaces under water. a** Atomic force microscopy (AFM) force curves indicating the deflection of the cantilever in the points marked on part (**b**). **b** AFM map of an infused Teflon-wrinkled surface under Milli-Q water ($c_{air} \sim 23.0 \pm 0.3$ mg kg$^{-1}$). The left panel shows the topography of the wrinkles; the middle shows the thickness of the lubricant and the right the air layer, as calculated by the analysis script (further details are contained in "Methods" and in the Supporting Information)[27]. The scale bars are in nm. **c** Cross-sections of the wrinkled surface before (left) and after (right) nanobubble nucleation, reconstructed along the dashed line from the profile in Supplementary Fig. S10a and part (**b**), respectively. Colors have been added to represent the different phases. **d** Laser confocal microscopy time-lapse of oil-infused Teflon wrinkles in the presence of highly gassed water with $c_{air} \sim 65 \pm 7$ mg kg$^{-1}$, showing the nucleation of a bubble over time. **e** Top view of bubble nucleation process over multiple sites. Scale bars are 25 μm. The full-time sequence is shown in Supplementary Fig. S9. Source data are provided as a Source Data file.

the nanobubble region before (left) and after (right) nucleation in Fig. 4c shows that the air pockets displace the lubricant locally, but the deeper regions between wrinkles retain silicone oil due to the large capillary forces. In the presence of air, the lubricant redistributes on the surface, increasing its thickness in other regions and further increasing slip in those areas. The constant redistribution of lubricant under different phases is expected[27]. Immersion of the LIS in gassed water ($c_{air} \sim 44 \pm 4$ mg kg$^{-1}$) induced an almost complete air layer (Supplementary Fig. S13). Although nanobubbles are well-known on flat hydrophobic surfaces[37], they have so far been ignored on LIS.

When LIS were immersed in highly gassed water ($c_{air} \sim 65 \pm 7$ mg kg$^{-1}$) and observed by confocal microscopy, microscopic air bubbles nucleated within seconds, as shown in Fig. 4d, e and Supplementary Fig. S9. Bubbles with a contact angle of 65 to 70° (in the air phase, consistent with the macroscopic contact angle of water on LIS in Table 1) nucleated extensively, grew over time, often exiting the field of view or coalescing with neighboring bubbles. The nucleation of bubbles is due to the air dissolved in the water and in the lubricant, diffusing and becoming pinned on exposed areas of the Teflon wrinkles.

Once formed, the nanobubbles are stable under the imposed experimental conditions. Following the approach by Samaha et al.[38], the terminal pressure at which air pockets should collapse from a Cassie to a Wenzel state is between 3.5 and 18 kPa (for an air-water interface with $\gamma_{wv} = 72$ mN m$^{-1}$, contact angle on flat Teflon of 120° and gas fraction of 0.9). However, under dynamic conditions of flow, the nanobubbles are expected to be cloaked by a thin film of silicone oil given the positive spreading parameter of silicone oil on a water-air interfaces: $S_{ow} = \gamma_{wv} - \gamma_{ov} - \gamma_{ow} = 10.5$ mN m$^{-1}$, where $\gamma_{wv}$, $\gamma_{ov}$, $\gamma_{ow}$ are the interfacial energies of the water-vapor, oil-vapor and oil-water interfaces, respectively[8,39,40]. The terminal pressure values for an air-oil interface ($\gamma_{wv} = 19$ mN m$^{-1}$) are between 1.8 and 9.5 kPa. Given that the maximum static pressure in our experiments is around 1.4 kPa (see Supplementary Fig. S4), it is expected that the air pockets will not collapse throughout the experiments.

The oil cloaking of the nanobubbles resembles the liquid-infused surfaces with entrapped air (LISTA) introduced by Hemeda and Tafreshi[41]. In LISTA, the lubricant is held within a double-re-entry solid geometry and recirculates on top of the entrapped air. For the configurations studied, LISTA present slip length values up to 37% higher than their LIS counterparts. In contrast with LISTA, in our system, the predicted cloaking lubricant layer on top of the bubbles is thinner (it could not be detected by the force measurements), due to negative disjoining pressure produced by the van der Waals interactions and hydrostatic pressure in the system (for this discussion see also ref. [40]). Compared to LISTA, the lubricant is not entrapped on top of the nanobubbles as it fully wets the solid substrate around them (see Fig. 4b). In addition, our substrate geometry is not re-entrant. Consequently, there is no opportunity for the lubricant to pin over the gas. This implies that the lubricant is free to flow over the bubbles, which increases the effective slip when compared with a recirculating case. As discussed later in this paper, it is possible that the lubricant layer can reduce the rate of air dissolution from the bubble into the working fluid, which increases the longevity of the bubble, as also suggested by Hemeda and Tafreshi[41].

Finally, the stability of the nanobubbles is confirmed by the fact that the drag reduction effect was maintained for more than 24 h, as shown in Supplementary Fig. S6. This is in agreement with previous studies. Nanobubbles have been found to be stable for days[42], and air pockets have been found to last up to 80 h in superhydrophobic surfaces underflow of water at low Reynolds numbers and low static pressures (<100 kPa) [43,44].

**Effect of nanobubbles on the effective slip**. The effect of surface bubbles on fluid slip has been widely studied[1,3,45]. Theoretical models predict that entrapped bubbles can produce both a slip and a no-slip condition, depending on the protrusion angle of the bubble (i.e., the angle from the horizontal at which the bubbles protrudes from the surface)[46,47]. Experiments by Steinberger et al.[48] confirmed these predictions using a surface containing a mattress of bubbles with a large protrusion angle: surface nanobubbles enhance slip only if the protrusion angle is small (typically smaller than 30°) and that the slip length is maximized when the angle is around 10°[49]. In our system, the nanobubbles nucleate within the surface roughness and seem to have a low protrusion angle, as shown in Fig. 4c. Therefore, the effect of the nanobubbles on the slip is twofold: they reduce the overall roughness of the wrinkled surface (similarly as the lubricant does) and provide an almost shear-free interface which significantly increases the local slip in comparison with the liquid lubricant counterpart.

The maps in Fig. 4b and SI demonstrate that LIS immersed in water are complex surfaces, consisting of areas infused with different fluid layers. As shown in Fig. 5a, the working fluid is exposed to specific local boundary conditions, determined by the distribution and thickness of the lubricating layers: (1) areas with nanoscale slip where the working fluid flows directly over the solid substrate (no-slip as measured in our microfluidic setup, grey label); (2) areas with low slip (hundreds of nm, yellow label), where only an oil lubricant layer is present; (3) areas with large microscale slip where nanobubbles are located on top of the solid substrate (purple label); and 4) areas with a large microscale slip where nanobubbles are located on top of the oil (orange label). By performing flow simulations over a surface with these mixed slip properties, we were able to obtain slip length values that quantitatively match the experimental values.

In Fig. 5b (left panel), the spatial distribution of the fluid layers for a LIS immersed in Milli-Q water is shown (derived from the map in Fig. 4b), with the colors representing the expected local boundary conditions. In this map, the majority of the area has low slip (yellow area), while the lower-left corner, where there is a nanobubble, is expected to display large slip. Using the thicknesses of the fluid layers measured by AFM, a local slip length was computed as a function of the coordinates $x$ and $y$ as shown in the central panel of Fig. 5b. The portions covered in the thicker nanobubble region display a maximum local slip length of 19 μm, while portions only with oil show a local slip smaller than 1 μm.

A three-dimensional numerical model was used to estimate the effective slip length across this surface. A Navier slip condition was applied on the bottom surface of the simulation domain based on the mapped slip length shown in the middle panel of Fig. 5b. The right panel of Fig. 5b shows the slip velocity on the bottom surface of the domain, which results in an effective slip of 1.1 μm. This theoretical value is close to the experimentally measured slip length of 3.8 ± 1.9 μm, measured with the same air content, shown in Fig. 1e. The calculated effective slip length is ten times larger than expected based on the presence of the lubricant alone, which explains our observed results.

As expected, the simulated $b_{eff}$ increases as the air coverage over the surface increases[18]. In Fig. 5c, the same processing and flow simulation was performed for a surface immersed in gassed water ($c_{air} \sim 44 \pm 4$ mg kg$^{-1}$, map shown in Supplementary Fig. S13). The left panel shows that a large portion of the surface is covered in the air (purple and orange labels in Fig. 5a). Compared with the previous case, the local maximum slip length is much larger (maximum 28 μm, central panel) and the effective slip length is estimated to be 5.2 μm (right panel), which is in excellent agreement with the experimentally measured value, shown in Fig. 1e.

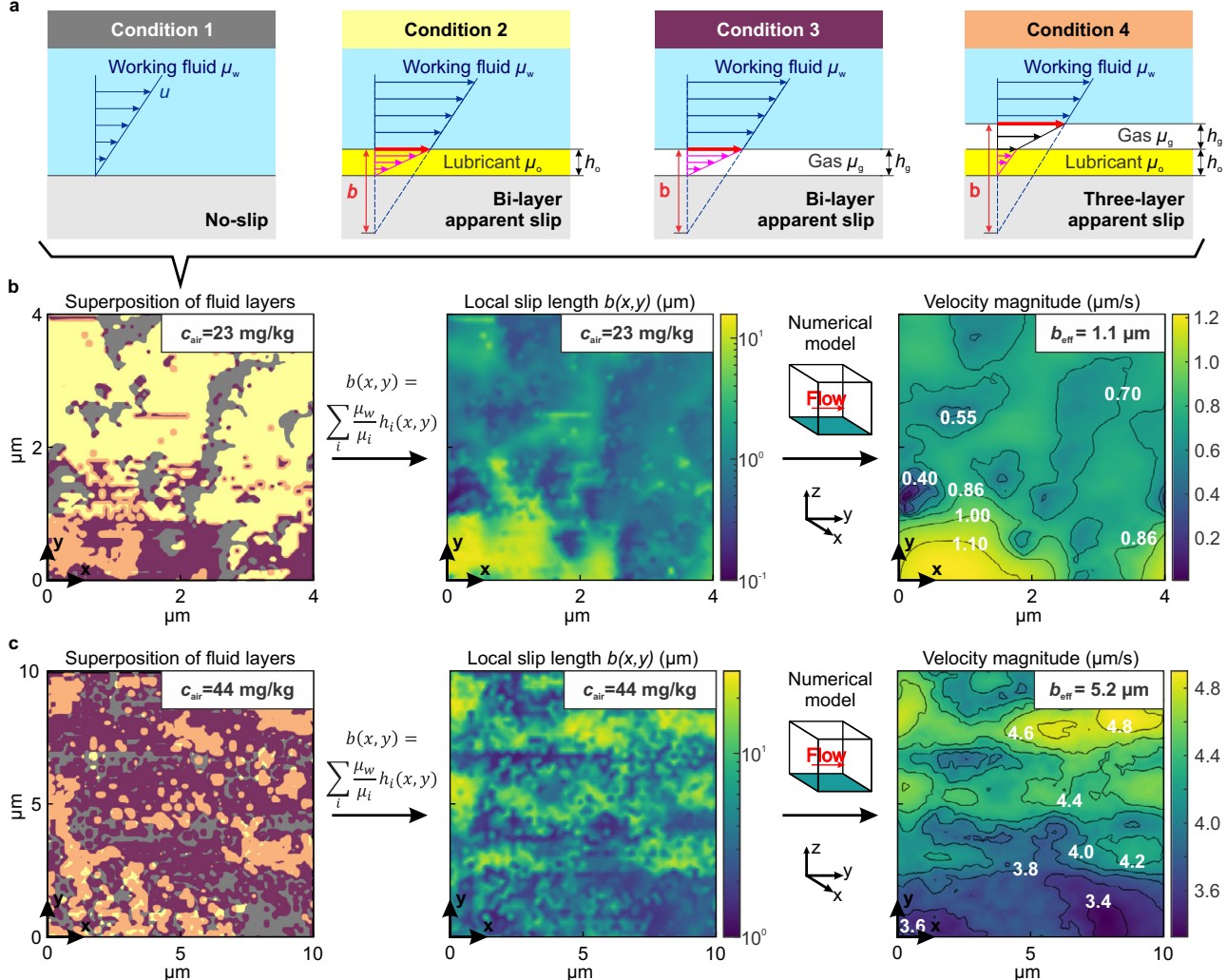

**Fig. 5 Fluid flow boundary condition and local slip on lubricant-infused surfaces. a** Schematic of the boundary conditions experienced by the working fluid on the solid surface on different lubricating fluid layers. **b** Left panel: spatial distribution of the lubricating fluid layers for a surface immersed in Milli-Q water with air content $c_{air} \sim 23.0 \pm 0.3$ mg kg$^{-1}$ (derived from atomic force microscopy map in Fig. 4b). The colors represent the flow conditions shown in (**a**). Central panel: map of the local slip length $b(x, y)$ estimated at a horizontal plane located at the highest peak of the wrinkled Teflon. $b(x, y)$ is computed as the sum of the apparent slip length given by each fluid layer according to Eq. (1), based on the lubricant layer thickness $h_o$ and $h_g$ measured by atomic force microscopy. Right panel: magnitude of fluid velocity immediately above the Teflon wrinkles, computed from a three-dimensional numerical model, for a tangential shear rate $\dot{\gamma} = 1$ s$^{-1}$ in the $y - direction$. Black lines are contour lines for the velocity. $b_{eff} = 1.1$ μm is the theoretical effective slip computed from the model. **c** Same panels as in (**b**), calculated for a surface immersed in gassed water in which $c_{air} \sim 44 \pm 4$ mg kg$^{-1}$ (atomic force microscopy map shown in Supplementary Fig. S13). Source data are provided as a Source Data file.

**Why do nanobubbles nucleate on LIS?** Despite extensive previous research on the plastron in superhydrophobic surfaces immersed in water[4,5,38,43,44,50], and on the stability of nanobubbles on smooth hydrophobic surfaces[37,51,52], nucleation of nanobubbles has not been reported in LIS. On the macroscopic scale, LIS create a smooth, liquid interface that eliminates pinning of the contact line for water droplets and most liquids[6]. Therefore the ability of LIS to trap and stabilize nanobubbles by pinning of the three-phase contact line has been so far overlooked. However, at the microscale, our AFM maps (Fig. 4c) show that the nanobubbles in LIS displace the lubricant. Therefore the pinning of the contact line occurs directly on the solid substrate. Air supersaturation of the working fluid is not necessary for nanobubble nucleation to occur[42]. Indeed low dissolved air content, as in typically employed Milli-Q water, was shown to lead to nucleation on the LIS (Fig. 4). Further, it has been reported that flow increases the rate of growth of nanobubbles on superhydrophobic

surfaces[53,54]. On LIS, higher flow rates increase the portions of exposed areas of the Teflon substrate to the water and, therefore, we expect that higher flow also increases nucleation site density. Our previous colloid probe AFM measurements[14,15] and our current microfluidic experiments on smooth infused surfaces did not show higher than expected slip length values. This corroborates the need for surface roughness for substantial air nucleation to occur in LIS.

The solubility of air in silicone oils is one order of magnitude higher than in water[55] and the diffusion coefficient is of the same order of magnitude as water. Therefore, lubricants can retain high air content and transport it from the working fluid to the underlying substrate. As a result, the presence of a lubricant layer does not impede the nucleation of nanobubbles.

In comparison with superhydrophobic surfaces, the LIS used here remain slippery and antifouling for long periods[23], yet localized portions of the substrate become exposed under

water[19,27]. Then the high roughness of the underlying hydrophobic wrinkles (roughness ratio of ~5)[23] enables the nucleation and stabilization of air pockets cloaked by a thin oil layer. The nucleation of air pockets is expected in LIS with similar high roughness and hydrophobicity.

It cannot be excluded that nanobubbles are present on the smooth OTS and PDMS substrates tested here, but no hydrodynamic drag reduction was observed. This could be explained in two ways: (1) they are not stable underflow given the surfaces' low roughness; (2) nanobubbles on OTS have a large contact angle[56]; therefore their geometrical configuration could cause low to no-slip as discussed above for protruding gas mattresses[45–48].

## Discussion

We have shown that lubricant-infused Teflon-nanowrinkled surfaces reduce hydrodynamic drag beyond expected values (up to 28%) under laminar flow conditions, even with lubricants more viscous than the flowing liquid (silicone oil 5 and 10 cSt, and hexadecane). The result is particularly striking for the flow of water (viscosity ratio $\mu_w/\mu_o = 0.1$): the apparent slip model would predict a slip length of about 100 nm, while we measure a value of $5 \pm 1\,\mu m$. The drag reduction remains unchanged for flows over 24 h. Our study has highlighted the discrepancy between the large slip observed in LIS and the apparent slip model. By studying the magnitude of slip relative to air content in the water, we explained its mechanism. We attribute the origin of the large slip observed in LIS to the nucleation of nanobubbles on the underlying substrate, which has not been reported before. The observed distribution of nanobubbles on the surface is sufficient to justify the large slip length measured. The ability to quantify the lubricant film spatial distribution on the nanoscale and map the presence of nanobubbles under water enables us to state these results confidently. Thus, we conclude that the presence of nanobubbles, well-established on other surfaces, is important even on infused surfaces, and can lead to counterintuitive phenomena such as slip in systems with low viscosity ratio.

We expect this mechanism to explain the high slip reported before on other rough LIS[10–12]. Apart from the fact that nanobubbles are difficult to dected[33], the main reason why their presence has been so far ignored on oil-infused surfaces is the assumption that as the lubricant is depleted from the surface, it is replaced directly by water. At the same time, we demonstrate that lost lubricant may be replaced by gas stabilized by the roughness in the underlying hydrophobic substrate. The Teflon wrinkles are not unique in this behavior. Still, we have chosen them because they are well-suited for studies of LIS underflow: (1) the winkles are nanostructured (the finest scale of wrinkle width and height is 200 nm), yet robust[22], and they trap lubricant well long-term (ref. [23] shows antifouling behavior is retained after over seven weeks of immersion in the ocean with only $0.2\,mL\,m^{-2}$ of lubricant); (2) they can be fabricated on the scale of several tens of $cm^2$ easily; (3) we can easily compare wrinkles in their superhydrophobic and LIS state; the antifouling behavior of wrinkled LIS is much superior to that of superhydrophobic wrinkles[23]; (4) we understand how the lubricant depletes from the wrinkles after immersion through a water/air interface, as a function of the chemistry of the lubricant[27].

From a fundamental point of view, these results have major implications for the theoretical framing of LIS immersed in water, and will impact our understanding of multiphase flows, flow through impregnated porous media, and oil extraction systems. For practical applications, this work will benefit drag reduction in micro- and nanofluidic devices, aid in the design of liquid-supported microfluidic devices, introduce new approaches to

fabricating and modeling immersed LIS and potentially lead to reassess the mechanism of other functions of LIS, such as antifouling.

## Methods

**Fabrication of tested surfaces.** Hydrophobized silicon wafers were produced using a standard procedure (immersion in a 3 mM octadecyltrichlorosilane, OTS, solution in toluene)[14]. Before silanization, silicon wafers were cleaned thoroughly: sonicated in ethanol and acetone, dried by nitrogen flow, plasma cleaned in air plasma (PDC-32G-2 Harrick Plasma). PDMS-grafted silicon wafers were produced following a recent protocol[21]. Briefly, clean silicon wafers were exposed to air plasma for 10 min, then left covered in silicone oil 350 cSt (200Fluid, Ajax Finechem) for 72 h at room temperature (22–25 °C), then rinsed by sonication in toluene and ethanol for 1 min in each solvent. Wrinkled Teflon substrates were produced by spin-coating (4000 rpm for 1 min) a 1.5% solution of Teflon AF in FC-40 over shrinkable polystyrene substrates (Polyshrink) and then annealed at 135 °C[22]. The substrates were infused by coating with oil (silicone oil 10 or 5 cSt, or hexadecane) overnight. The excess lubricant was drained by placing the substrates vertical for 1 h prior to being tested[23]. Contact angle measurements were made using a KSV CAM 200 goniometer.

**Control over air content in working fluids.** Water with different air content was used in the experiments: degassed water, Milli-Q water as produced, gassed water and highly gassed water (Supplementary Tables S4 and S5). The oxygen concentration in the fluid was measured using a dissolved oxygen sensor (RCYACO, Model DO9100), and its value was used to estimate the air concentration in the liquid. Milli-Q water as produced was measured to be effectively air-saturated at atmospheric pressure (101 kPa) and had an air content of $c_{air} \sim 23.0 \pm 0.3\,mg\,kg^{-1}$ (average of six measurements and standard deviation). In order to change the air content in the water, 20 mL of Milli-Q water were placed under a pressurized atmosphere of air for at least 30 min and up to 4 h in a 50-mL Falcon tube placed horizontally (see the setup in Supplementary Fig. S2). The sample was shaken at intervals of approximately 10 min which helps with the mixing and air saturation of the liquid. Air pressures of 6, 203, and 304 kPa were used, which produced an air content of $c_{air} \sim 1.4 \pm 0.5$, $44 \pm 4$, and $65 \pm 7\,mg\,kg^{-1}$, respectively. For pressures of 203 and 304 kPa, it was found that in the first 30 min of pressurization, the air content in the liquid reaches at least 80% of the saturation value given by Henry's law. The saturation then increases up to 90% after one hour. On the other hand, for water placed under 6 kPa, the oxygen level was reduced from 8.8 to 0.7 mg kg⁻¹. More details are provided in "Methods" and the Supporting Information. Highly gassed water (~65 ± 7 mg kg⁻¹) could not be used in the pressure drop microfluidic experiments due to the clogging of the channel by large air bubbles.

The fact that large slip was still evident in degassed water has at least two explanations. First, it is difficult to completely remove gas from water using conventional degassing procedures[57], as quantified using the oxygen sensor. Second, bubbles could be formed when first filling the microfluidic channel with water, consistent with our observation that the pressure drop reduction remained constant throughout our measurements. This effect has been observed before. For example, Watanabe et al. found that the drag reduction of superhydrophobic surfaces did not change when using saturated or degassed water[58].

**Pressure drop vs flow rate experiments.** Two custom-built microfluidic devices were used, one made of PMMA and the other of aluminum. The geometry and dimensions of these devices are presented in Supplementary Fig. S1. The density, viscosity and interfacial tension of the fluids used in these experiments were characterized using a density meter (DMA 35N, Anton Paar), a modular compact rheometer (MCR 302, Anton Paar) and a tensiometer CAM 200 (KVS Instruments), Supplementary Table S1. During the pressure drop vs flow rate experiments, the working fluid flow rate was accurately measured using a flow sensor (MFS3 or MFS4, Elveflow) at the inlet port. The pressure drop along the channel was measured using two pressure sensors (MPS0, Elveflow) with an accuracy of 20 Pa or a differential pressure sensor (PX459-10WDWU5V, Omega) with an accuracy of 2 Pa. The experimental uncertainty in the estimation of $\Delta p_{no-slip}$ was minimized as described in "Methods" and the Supporting Information and in ref. [20].

For water, the flow rates used correspond to Reynolds number of $\mathcal{O}(1)$ and capillary number of $\mathcal{O}(10^{-4})$ (except for the flow rate of 800 μL min⁻¹ which correspond to a $Ca = 0.001$). With different working fluids, the flow rate was adjusted to keep a constant $Ca = 0.001$ in all experiments. Before the start of measurements, each channel was flushed for 10 min at 200 μL min⁻¹ in order to remove excess oil on the substrate. Supplementary Table S2 shows the number of repetitions carried out for each experiment.

**Laser confocal scanning microscopy.** Laser confocal scanning microscopy was used to study (1) the stability of the lubricant film on Teflon wrinkles underflow and (2) the nucleation of gas bubbles on the LIS. For the first experiment, the infused layer of silicone oil (stained with Nile red at ~0.1 mM) was imaged underflow of degassed water (not dyed) using an Olympus FluoView FV3000

confocal microscope with a 488 nm laser and a ×60 objective, see "Methods" and Supporting Information and Supplementary Fig. S7. For the study of gas nucleation on the infused Teflon wrinkles (silicone oil 10 cSt, not dyed), the water was dyed with a low concentration of acridine orange and excited with a 488 nm laser using a ×40 silicone oil objective. Here, in order to nucleate observable micrometric bubbles, water with high air content was used (~65 ± 7 mg kg$^{-1}$). The appearance of microbubbles was taken as evidence of the high affinity of air for the infused wrinkled substrate, see Supplementary Fig. S9. The signal in reflection from the wrinkled substrates is not sufficiently resolved to distinguish a gas layer before nucleation of a micro-bubble.

**Lubricant film mapping using meniscus force atomic force microscopy.** AFM meniscus force measurements, performed in air using a MFP-3D AFM (Asylum Research), were used to quantify the nanoscale thickness of silicone oil before and after exposing the surface to flow for 30 min[19]. AFM cantilevers with spring constant 1–7–N/m (Multi75, Budget Sensors, Sofia) were used. The average lubricant film thickness before shearing was found to be 0.8 ± 0.2 μm on three different samples, which can be extrapolated to a volume of lubricant per channel of ~80 nL. Excess lubricant droplets are present on the surface in air, and they spread fully when the surface is moved underwater, increasing the lubricant level by a few hundred nm[27]. Therefore, underwater the lubricant thickness should be of the order of 1 μm corresponding to a total volume of lubricant per channel of ~100 nL, see "Methods" and Supporting Information and Supplementary Fig. S8. The lubricant thickness could be overestimated if lubricant accumulated on the AFM tip. However, repeated experiments suggest that this is not an issue: when a clean dry sample was mapped directly after an infused one, lubricant layers were not observed; samples were scanned with more than one tip and the measured thickness values averaged over several scans.

**AFM mapping of nanobubbles on oil-infused Teflon wrinkles.** AFM meniscus force curves were used to reveal air pockets underwater, using a method developed by us[33]. Nanobubbles were identified by the characteristic positive deflection (i.e., repulsion) due to the air/water interface[34,35,59–61]. Underwater force maps were collected on silicone oil-infused Teflon wrinkles in a custom AFM cell. The AFM tip was hydrophobized via deposition of a thin layer of PDMS (~1.4 nm).

Force curves were analyzed using a Python script to automatically detect the presence of either an air layer, a lubricant, or an air layer on top of a lubricant layer. The script does this by first finding all regions of the force curve with a rapid change in gradient by finding peaks in the second derivative. It then looks at the portion of the force curve between these peaks and determines whether the shape resembles that of a lubricant layer or an air layer. An air layer is defined by having at least 60% of the points having a positive gradient, and the overall change in deflection must be less than a predetermined threshold (0.5 nm in this study). The accuracy of this technique was verified by mapping the partially collapsed plastron on superhydrophobic (non-infused) Teflon wrinkles underwater, shown in Supplementary Fig. S12. Details about this technique can be found in "Methods" and in the Supporting Information. The code used for processing the data is available online (refer to ref. [62]).

**Spatially dependent slip length and numerical computation of the effective slip length.** AFM maps demonstrate that our LIS are composite, made of a patchwork of areas of exposed solid surface, oil film, and gaseous layer. Therefore, the slip length varies at different locations, as shown in Fig. 5. Figure 5b, c (left panels) was obtained using Matlab-processed AFM data to superimpose the fluid layers and represent areas with different boundary conditions. The gray color corresponds to regions with no oil and no air. Yellow areas represent the locations with oil thickness greater than 5 nm, and purple and orange regions correspond to air bubbles of thickness larger than 5 nm.

The corresponding local slip length presented in Fig. 5b, c (central panels) was estimated by adding the effect of each fluid layer using the apparent slip model (Eq. (1)) for an imaginary horizontal plane crossing the highest peak of the surface topography. The local slip length includes the contribution of the lubricant layer, gas layer and working fluid up to that plane.

The local slip was then mapped as a Navier slip length to the bottom wall of a three-dimensional CFD domain in COMSOL Multiphysics. The domain is a prism of square base $S \times S$ in the $xy-plane$, with $S$ the size of the AFM map shown in Fig. 5, and $H = 10\,\mu m$ the height in $z-direction$. A Couette flow is induced in the domain by imposing a tangential shear rate $\dot{\gamma} = 1\,s^{-1}$ in the upper wall of the domain in the $y-direction$. The two lateral walls of the domain were defined as inlet and outlet, respectively, while the remaining two lateral walls were defined as symmetry planes. The fluid filling the whole domain was water. Then, the effective slip was estimated as:

$$b_{\text{eff}} = \frac{\overline{u}(H)}{\dot{\gamma}} - H, \tag{5}$$

where

$$\overline{u}(z) = \dot{\gamma}(z + b_{\text{eff}}), \tag{6}$$

is the average velocity as a function of $z$ assuming a linear velocity profile with an additional slip $b_{\text{eff}}$.

A refined mesh was used to guarantee that the results are mesh-independent. In addition, the estimated effective slip seems neither to vary significantly with the orientation of the mapped slip length on the bottom wall nor with the change of $H$ for cases in which $H > S$.

## Data availability
The raw and processed data generated in this study have been deposited in the Figshare database under accession code 17096963 https://figshare.com/articles/dataset/Source_data_zip/17096963. Source data are provided with this paper.

## Code availability
The code used to process the atomic force microscopy data is available in https://github.com/speppou/AFM_Nanobubble_Mapping.

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

## Acknowledgements

C.N. acknowledges the Australian Research Council for funding (FT180100214). The authors acknowledge the facilities and assistance at the Australian Centre for Microscopy & Microanalysis at the University of Sydney. C.V.S. thanks the Costa Rican Ministry of Science and Technology for funding and Tecnológico de Costa Rica for computational resources. S.P.C. acknowledges the Australian Government Research Training Program (RTP) Scholarship for support. The authors thank Prof. Ronald Larson, Prof Hans-Jürgen Butt, Prof. Steve Armfield, and Prof. Maryanne Large for useful discussions.

## Author contributions

C.V.S. and C.N. designed research; C.V.S., S.P.C., and L.Z. performed experiments; C.V.S. and C.N. developed numerical model; S.P.C. performed meniscus force AFM and created Python script for processing data; C.V.S., S.P.C., L.Z., and C.N. analyzed the data; C.V.S., S.P.C., L.Z., and C.N. wrote the paper.

## Competing interests

The authors declare no competing interests.
