## [Peer Review File · Nature Communications]

REVIEWER COMMENTS

Reviewer #1 (Remarks to the Author):

The manuscript by Sanchez et al. reports significant slip effect on a submerged lubricant-infused surface made of Teflon with nano-wrinkles. The authors attribute the effect to the nucleation and growth of nano-bubble on their surface during the experiment. This referee finds the observations interesting but feels that, beyond a series of microscopic images and AFM test data, the work lacks a solid scientific foundation.

Specific Concerns:

1-It is not clear what exact scientific discovery is reported here. The authors attributed the slip effect to the nucleation of air bubbles on their wrinkled Teflon surface after it was infused with oil. Was this due to the presence of oil or due to the wrinkles? Would the authors not see the same (or better) effects with a lubricant-infused nanofabricated surface? What is the uniqueness of wrinkled Teflon or the oils used in the experiments? What is the underlying physics of this observation?

2-The reported slip effect is solely due to the presence of air nanobubbles (the lubricant cannot produce much slip according to the LIS literature). Therefore, all that discussion in the text about "mechanisms for slip length" seems to be redundant.

3-The slip effect on a submerged superhydrophobic surface is due to the presence of air, and the air is proven to dissolve and disappear in water over time, and in the presence of flow (see the papers by Tafreshi and Gad-el-Hak and their co-workers). The authors should calculate the longevity of the nanobubbles trapped in their wrinkled surface to present a scientific basis for their claims (e.g., that the nanobubbles are stable over time and/or they grow with increasing the flow velocity). Generally speaking, the authors do not seem to present any results with regards to time-independence of their slip effect measurements, which is a concern.

4-The results obtained from the authors' numerical simulations are not useful. These simulations were conducted for two-phase flow systems (oil and water) while the entire effect is due to the presence of air, and so a three-phase flow simulation is needed. The entire discussion about these water-oil simulations and why they do not match the experiment is trivial. Also, claiming that a recirculating flow is created in a film of oil with a thickness as thin as 1 micron needs a better proof than the simple steady-state simulations given in the manuscript.

5-In a very short but yet intriguing statement, the authors mentioned about the possibility of oil cloaking the nanobubbles. This is a very interesting hypothesis (perhaps more interesting than the rest of the paper) but the authors did not elaborate on it at all (e.g., not included in Figure 5a?!). If this happens to be the case, then the authors need to also explain how the oil-cloak allows a recirculating flow to form inside the air bubble (see the 2016 Langmuir paper by Hemeda and Tafreshi).

6-In a self-contradicting statement the authors mentioned about the importance of pinning for stabilizing the air bubbles, but they also stated that their slippery LIS surface prevents pinning. If pinning is important, then why the oil was needed? In fact, the authors never presented a clear scientific reason as to why the oil infusion was needed (something beyond what was or was not observed).

7-Another controversial statement is made before the Discussion section: "...lubricants can retain high air content and they can also transport it from the working fluid to the underlying substrate. As a result, the presence of a lubricant layer is not an impediment for the nucleation of nanobubbles and may even promote it." If the air was not cloaked by the oil, then why could the oil layer be an impediment for air bubble nucleation?

In summary, this is an interesting manuscript but a much better scientific basis is needed for the claims made in the manuscript. As mentioned earlier, the manuscript is filled with AFM images (technology demonstration from authors' past publications) but is short in presenting a theoretical basis for the results. Figures 4a and 4b, for instance, could easily be moved to the SI with no harm to the paper. The manuscript should be revised and re-reviewed.

Reviewer #2 (Remarks to the Author):

Recently, nanobubbles have become a hot topic in interfacial science due to their unique properties and potential applications. In this study, Vega-Sanchez et al. investigated the drag reduction effect of surface nanobubbles on LIS. Although there have been many reports on the use of superhydrophobic surfaces to reduce drag in hydrodynamics, the results in this study show that nanobubbles can achieve the same drag reduction effect. The authors measured the drag reduction of nanobubbles on different surfaces; it is found that the drag reduction effect is related to the gas

concentration in the solution. Furthermore, force spectroscopy experiments were used to characterize the nanobubbles in different systems, and the numerical simulation was used to verify the conclusion.

In most of the previous researches, micrometer-sized bubbles trapped at superhydrophobic surfaces were used for drag reduction. However, these microbubbles are not stable, under sufficient pressure or shear stress, the microbubbles may release or dissolve in the solution, and the drag reduction effect may disappear. According to the data reported in this study, nanobubbles may be more stable and their drag reduction effect is not affected. If the drag reduction effect of nanobubbles can be used in hydrodynamics, many applications are expected. The reviewer supports the publication of this manuscript in Nature Communication and would like to ask the authors to address the following questions:

1. Please discuss in detail the advantages of using nanobubbles on LIS to achieve drag reduction compared with the traditional superhydrophobic surfaces. These contents will increase the impact of the article.

2. In the experimental method, the author described how to control the concentration of gas in water. By placing water in a chamber that containing gas with a certain pressure, until it reached an equilibrium state. It is worth noting that it takes a long time for the gas molecules to diffuse into the water to reach equilibrium when the water is at rest. 30 mins may not be enough for them to reach equilibrium. According to the reviewer's experience, the concentration of the gas will only reach 90% of the equilibrium value overnight. So, if possible, consider using an oxygen meter to monitor the gas concentration in the solution. Authors can refer to methods in the reference.¹

3. In some Figures, the authors give the AFM images in the same position under different conditions. Please briefly introduce how to repositioning in different experiments.

4. Please explain why the drag reduction is still obvious when using degassed water, as shown in Figure 3. Because nanobubbles should dissolve in degassed water. Does the TW-air surface have drag reduction when degassed water is used?

5. Although it is generally believed that the bubble surface cannot sustain shear stress, some experiments show that this boundary condition is very sensitive to the adsorption on the gas-liquid interface. When a small amount of surfactant (or contaminants) is adsorbed on the gas-liquid interface, the gas-liquid interface may become no-slip². Please briefly comment on the influence of this effect on the experimental results.

1. M. A. Borden and M. L. Longo, *Langmuir*, 2002, 18, 9225-9233.

2. P. Pawliszak, V. Ulaganathan, B. H. Bradshaw-Hajek, R. Manica, D. A. Beattie and M. Krasowska, *The Journal of Physical Chemistry C*, 2019, 123, 15131-15138.

Reviewer #3 (Remarks to the Author):

Vega-Sánchez et al. reported that the large slip on infused Teflon wrinkled surfaces could be explained by the formation of nanobubbles. The study is significant for the transportation of fluid. They also gave an evidence on the production of air layer and large bubbles, such as their tomography and force measurements. Actually, the idea is not new anymore. Many research presented the existence of nanoscale bubbles or gas layers would increase the slip length since surface nanobubbles were observed by AFM in 2000 (Nature Materials 2, 221-227(2003); Langmuir 18, 3413-3414 (2002); Physical Review E 70, 026311(2004); Nanotechnology, 2009, 20, 045301; Soft Matter, 6, 29-66(2010); J. Vac. Sci. Technol. B 18(5), 2573-2575(2000)). This study reported more detailed experiments to prove the existence of surface nanobubbles or gas absorption would increase the slippage of LIS. Generally, gas nanobubbles are easily formed on hydrophobic surface. I am wondering that they did not find evident slip change on OTS coated silicon wafer. It is suggested that authors should measure the results of slip on OTS wafer with a large content of air in water. Dissolved oxygen concentration may more precise than estimated by Henry's law. Please check the Figure 3c in page 5, "blue circle in Fig.3c" or Fig.3e? Also, ...as a dashed line in Fig.1 and Fig.3c..., here Fig.3c or Fig.3e?

Reply to reviewer's comments

Reviewer #1:

Reviewer comment:

1-It is not clear what exact scientific discovery is reported here. The authors attributed the slip effect to the nucleation of air bubbles on their wrinkled Teflon surface after it was infused with oil. Was this due to the presence of oil or due to the wrinkles? Would the authors not see the same (or better) effects with a lubricant-infused nanofabricated surface? What is the uniqueness of wrinkled Teflon or the oils used in the experiments? What is the underlying physics of this observation?

Answer:

High slip is well known in liquid infused surfaces, however, remarkably no attempt to understand the origin of this high slip has been made. Our significant contribution is to explain why liquid infused surfaces are so slippery. We focus on liquid infused surfaces as these are important for anti-fouling applications, where they are more effective than superhydrophobic surfaces. We show that the high slip occurs because, unexpectedly, the lubricant is replaced by air. While we have chosen to use Teflon nanowrinkles, we think this effect is general to other liquid infused surfaces (ref [10]-[12]), as it provides a clear explanation for all the high slip lengths observed in these liquid infused systems. The underlying superhydrophobic surface (Teflon nanowrinkles) is preferably wetted by air than water, so when the lubricant is lost it is replaced by air. This replacement of lubricant by air only happens in isolated microscale spots on the surface, but this is sufficient to dramatically reduce drag.

This comment suggests to us that the main aim of our paper needs to be clarified, and we have done this on pages 1, 3, 5 and 9. We have accentuated that our main aim is to investigate the mechanism underpinning the unexpectedly large (microscale) interfacial slip observed on lubricant-infused surfaces (LIS).

The new underlying physics we reveal is that the slip mechanism on LIS is more similar to that on superhydrophobic surfaces (SHS) than previously appreciated. It is already well known that gas nucleates in the form of a plastron on immersed SHS, including on our plain (uninfused) wrinkles (ref [21]). Bubbles also nucleate naturally on LIS when portions of the solid surface are exposed to the water. This mechanism in LIS has not been considered before, as: 1) the assumption is that when the oil is depleted, water (the flowing liquid) immediately fills the gaps; 2) acquiring experimental evidence of nanobubbles (particularly on a structured surface) is complex as they are not overt like the plastron on SHS. High surface roughness appears to be needed for the large slip effect to be observed, as only nanoscale slip was found on smooth hydrophobic surfaces (either infused or non-infused,

ref. [14] and [15]), even though in principle some nanobubbles could appear on smooth hydrophobic surfaces too.

The gas nucleation process is not unique to our infused wrinkled Teflon surfaces and is expected to occur in all LIS with similar characteristics (rough and hydrophobic surfaces) where large slip has been observed but not explained - such as the highly rough and hydrophobic substrates in Solomon et al. (Langmuir, 2014), Kim and Rothstein (Exp. Fluids, 2016), and Sang and Lee (Soft Matter 2019). We have clarified this on pages 1 and 3. There is nothing unique about the silicone oil, and indeed hexadecane produces the same large slip effect. What is needed for stable LIS is a lubricant that spreads fully on the structured surfaces when immersed under water.

Currently, slip on LIS is explained by using a fluid mechanics approach of a two-phase fluid system (e.g. water and oil). Both our results with oil-infused wrinkled surfaces and those with previously published LIS can only be explained with a three-phase fluid system. The results on the underlying superhydrophobic Teflon wrinkles (prior to infusion, Fig 3b and d, blue stars) are well in agreement with the expected values of slip, based on the theory by Ybert et al. (*Phys. Fluids* **19**, 123601 (2007)). We have clarified this on page 5, by including comparison with Ybert et al's theory. We hope that with this and our other additions our main aim is now clearer.

Finally, we have chosen the wrinkled surfaces for this study because they are well-suited for studies of LIS under flow: 1) Our wrinkles are nanostructured (the finest scale of wrinkle width and height is 200 nm), yet robust (see ref. [21]), and they trap lubricant well long-term (ref. [22] shows anti-fouling behavior is retained after over seven weeks of immersion in the ocean with only 0.2 ml/m² of lubricant); 2) they can be fabricated on the scale of several tens of cm easily; 3) we can easily compare wrinkles in their SHS and LIS state; the anti-fouling behavior of wrinkled LIS is much superior to that of wrinkled SHS; 4) we have quantified how the lubricant is depleted from the wrinkles after immersion through a water/air interface, as a function of the chemistry of the lubricant (ref. [19] and [26]).

Reviewer comment:

2-The reported slip effect is solely due to the presence of air nanobubbles (the lubricant cannot produce much slip according to the LIS literature). Therefore, all that discussion in the text about "mechanisms for slip length" seems to be redundant.

Answer: We disagree with the reviewer's reading of the literature on LIS. The well-accepted mechanism of fluid slip on LIS is that the presence of the liquid lubricant alone explains the large slip measured in LIS. We reveal that this is not the case, with the discrepancy between model and experiments overlooked so far. We proceed to explain the effect directly using experimental measurements of nanobubbles (thickness and distribution). It is highly satisfying that the observed distribution of nanobubbles quantitatively explains the large observed slip length values. We have clarified this in page 3.

Reviewer comment:

3-The slip effect on a submerged superhydrophobic surface is due to the presence of air, and the air is proven to dissolve and disappear in water over time, and in the presence of flow (see the papers by Tafreshi and Gad-el-Hak and their co-workers). The authors should calculate the longevity of the nanobubbles trapped in their wrinkled surface to present a scientific basis for their claims (e.g., that the nanobubbles are stable over time and/or they grow with increasing the flow velocity). Generally speaking, the authors do not seem to present any results with regards to time-independence of their slip effect measurements, which is a concern.

Answer: We provided time-dependent measurements (Fig. S6 in new manuscript), showing that the drag reduction effect, due to bubbles, is stable for 24 hours. Extensive work by Xuehua Zhang and others (e.g. Seddon et al. ChemPhysChem 13, 2012) reported that nanobubbles on smooth surfaces can be stable for days. Although the main aim of our study is not the longevity of the drag-reducing properties of LIS, we have added calculations showing that our results are in good agreement with the work by Tafreshi and Gad-el-Hak mentioned by the reviewer, as detailed below. We have added this information to the manuscript in page 7.

- The collapse of the trapped air due to hydrostatic pressure is not a concern in our experiments. Following the approach presented by Samaha, Tafreshi and Gad-el-Hak (Phys. Fluids 23, 2011), we estimated that the terminal pressure at which the superhydrophobic surface collapses from a Cassie to a Wenzel state is between 3.5 and 18 kPa (based on wrinkle topography shown in Fig. S5, contact angle on flat Teflon of 120° and gas fraction of 0.9). The values for the same calculation but assuming an air/oil interface, instead of pure air/water, is 1.8 and 9.5 kPa. Given that the maximum static pressure in our experiments is around 1.4 kPa (see Fig. S4), it is expected that the air pockets will not collapse throughout the experiments.
- The lifespan of gas pockets on micro/nanoscale superhydrophobic fiber surfaces was found to be approximately 80 hours and 30 hours for a static pressure of 0 kPa and 200 kPa, respectively, even in undersaturated water (Samaha, Tafreshi, Gad-el-Hak, Phys. Fluids 24, 2012). The maximum pressure to achieve a fully collapsed Wenzel state on their micro/nanoscale surfaces (inter-fiber distance between 5 μm and hundreds of nanometers) was estimated to be two orders of magnitude higher than that required for surfaces with microscale roughness (100 μm). Our wrinkled Teflon surfaces are nanoscale (spaced apart ~ 180 nm) on top of larger scale features (2.5 to 13 μm , as shown in Fig. S5), and many of our experiments were carried out with gassed water, therefore a similarly long lifespan of the gas pockets is fully expected.
- The longevity of the gas layer on the same superhydrophobic surfaces under flow was found to be 75 h, 15 h and 14 h for Reynolds numbers of 0, 997 and 6023, respectively (Samaha, Tafreshi, Gad-el-Hak, Langmuir 28, 2012). Our experiments were carried out at Reynolds numbers smaller than 11, therefore the longevity of our gas pockets is expected to be in the range of many hours as well. In contrast to the experiments reported by Samaha et al., we used gassed water in many of our experiments, which promotes nucleation of gas on the surface, as shown in Fig. S11 and S13).

Reviewer comment:

4-The results obtained from the authors' numerical simulations are not useful. These simulations were conducted for two-phase flow systems (oil and water) while the entire effect is due to the presence of air, and so a three-phase flow simulation is needed. The entire discussion about these water-oil simulations and why they do not match the experiment is trivial. Also, claiming that a recirculating flow is created in a film of oil with a thickness as thin as 1 micron needs a better proof than the simple steady-state simulations given in the manuscript.

Answer: Two numerical simulations are presented in the manuscript. One is a 2D two-phase flow (Fig. S15 in new manuscript) and the other one is a 3D single-phase flow (Fig. 5). The former demonstrates that the presence of lubricant alone cannot explain the slip measured. We agree with the reviewer that this water-oil simulation is not strictly necessary, but it dispels any questions about the potential effect of the recirculation cavities, a common explanation for the observed large slip. On the contrary, the 3D simulation is enabled by data on the three-phase system, as it includes boundary conditions obtained from our maps of oil and gas thicknesses. For example, in Fig. 5b, the maximum local slip length, measured directly on top of the nanobubble of 100 nm thickness, is 19 μm , but the 3D simulation reveals that the effective slip over the whole area, including regions with oil only, is only 1.1 μm . So, the 3D simulations use boundary conditions from the three-phase system and are crucial to correlate the presence of nanobubbles with the micrometric slip measured in our LIS.

Finally, although recirculating flows are insufficient to explain any of our results, they are expected to occur even at smaller scales based on the literature. There is plenty of evidence that fluids flowing through spaces larger than a few tens of nanometers can be treated as a continuum medium (e.g., Cheng & Giordano, Physical Review E, 2002 used channels of radius 40 nm to 2.7 μm). Therefore, we assume our oil film of 1 μm thickness can be treated as a continuum. The shear stress and velocity of the lubricant should match the ones of the working fluid at the interface. If the lubricant is in a confined space (e.g., the cavities produced by the wrinkled Teflon), it is forced to recirculate, otherwise a discontinuity of the velocity and shear stress at the interface would occur. Additionally, the lubricant is retained in the surface topography even after being exposed to flow for at least 30 min (see Fig. S7 and Fig. S8 in new manuscript). In the absence of nanobubbles, this shows that the lubricant recirculates within the surface topography and, when we have presented these results at conferences, it has been suggested that recirculation could be an explanation for the large slip observed (it is not).

Reviewer comment:

5-In a very short but yet intriguing statement, the authors mentioned about the possibility of oil cloaking the nanobubbles. This is a very interesting hypothesis (perhaps more interesting than the rest of the paper) but the authors did not elaborate on it at all (e.g., not included in Figure 5a?!). If this happens to be the case, then the authors need to also explain how the oil-cloak allows a recirculating flow to form inside the air bubble (see the 2016 Langmuir paper by Hemeda and Tafreshi).

Answer: Cloaking of the bubbles is a peripheral aspect of the paper, and none of our conclusions depend on the cloaking layer. Oil cloaking of water droplets in air is not a controversial topic and has been reported in many papers (starting from Smith et al. *Soft Matter* 2013 onwards; many examples of oil cloaking air can be found in our recent review: Peppou-Chapman et al., *Chem Soc Rev*, 2020, 49, 3688–3715). Silicone oil cloaking of a water-air interfaces is expected to occur because the spreading parameter S_{ow} of the oil over the water is positive: $S_{ow} = \gamma_{wv} - \gamma_{ov} - \gamma_{ow} = 10.5 \frac{mN}{m}$, where $\gamma_{wv}, \gamma_{ov}, \gamma_{ow}$ are the interfacial energies of the water-vapor, oil-vapor and oil-water interfaces, respectively. Kreder et al. (*Physical Review X*, 8, 2018) showed that under dynamic conditions the oil cloaking always occurs, the thickness is between a few hundred and few tens of nm, and the lubricant film thickness is not uniform. We have added a sentence in page 7 on this.

In the computational work by Hemeda & Tafreshi (*Langmuir*, 32, 2016) a three-phase system consisting of working fluid, oil and air is presented. However, there the oil layer is suspended over an air layer, pinned, and confined between two re-entrant solid features. This pinning situation is unlikely to occur on our wrinkled Teflon surface, because the silicone oil fully spreads on Teflon under water. In our work, the wrinkles were infused with lubricant prior being exposed to flow of water or glycerol-water mixtures. Therefore, it is expected that the nm-thin lubricant film formed on top of a nanobubble will flow with the external fluid and not necessarily recirculate on top of the bubble. Further thinning of the lubricant film is expected due to static pressure as well. These are rational hypotheses, however, this is not a simple matter to resolve, as revealed in Kreder's paper, and should be addressed in a separate, dedicated study, for example using combined interference microscopy and flow.

Reviewer comment:

6-In a self-contradicting statement the authors mentioned about the importance of pinning for stabilizing the air bubbles, but they also stated that their slippery LIS surface prevents pinning. If pinning is important, then why the oil was needed? In fact, the authors never presented a clear scientific reason as to why the oil infusion was needed (something beyond what was or was not observed).

Answer: Again, this comment highlights that a clarification was needed, and we have modified the introduction of our paper to deliver our aim fully, as discussed in reply to point #1. Our main aim is to investigate the mechanism of the unexpectedly large interfacial slip on lubricant-infused surfaces. The oil is needed to produce LIS.

Regarding the pinning of the contact line on LIS, two cases should be distinguished: the pinning of water droplets at the macroscopic scale and the pinning of nanobubbles at the microscopic scale within the structure of the LIS. LIS are generally described as low adhesion surfaces due to the way that macroscopic water droplets move with low contact angle hysteresis on them due to the absence of interaction between the water and the underlying solid (the droplets oleoplane; see Daniel et. al 2018). In this case, pinning refers to the water/air contact line not pinning on the solid substrate, and, in this case, the length scale of the contact line is orders of magnitude greater than surface roughness. In our case, silicone oil fully spreads on the Teflon underwater (spreading coefficient $S > 0$) and, therefore, a water droplet does not contact the underlying solid substrate. The pinning we describe in

the paper is on the nanoscale and involves the pinning of the nanobubble on the solid substrate. Here, silicone oil dewets from the solid substrate in air in our LIS (spreading coefficient $S < 0$). Again, this involves a water/air contact line, but as the nanobubble nucleates at the solid substrate, it can displace the oil and the three-phase contact line will contact the solid. Additionally, the length scale of the nanobubble is on the same order as the roughness, aiding in the stability. A detail explanation of this is presented in Smith et al. Soft Matter (2013). We have added a clarification on page 9.

Reviewer comment:

7-Another controversial statement is made before the Discussion section: "...lubricants can retain high air content and they can also transport it from the working fluid to the underlying substrate. As a result, the presence of a lubricant layer is not an impediment for the nucleation of nanobubbles and may even promote it." If the air was not cloaked by the oil, then why could the oil layer be an impediment for air bubble nucleation?

Answer: There is no report in the literature on the nucleation of gas bubbles on oil-infused surfaces (any type of LIS). The assumption is that, as the lubricant is depleted from the surface, it is replaced directly by water (or flowing liquid), ignoring the possibility that it could be replaced by gas. Our measurements are the first to reveal this new mechanism, hence this statement seems justified. Readers might be aware of literature reporting slip being decreased or eliminated by the presence of trace contaminants (as in point #5 by reviewer 2), which is clearly not occurring here in presence of silicone oil or hexadecane.

Reviewer comment:

In summary, this is an interesting manuscript but a much better scientific basis is needed for the claims made in the manuscript. As mentioned earlier, the manuscript is filled with AFM images (technology demonstration from authors' past publications) but is short in presenting a theoretical basis for the results. Figures 4a and 4b, for instance, could easily be moved to the SI with no harm to the paper. The manuscript should be revised and re-reviewed.

Answer This paper has revealed and investigated in detail the mechanism by which LIS reduce drag, which is of significance to wetting, interfacial science and microfluidics. We think that this new insight could lead to many new studies on multiphase flow, flow in porous media and oil extraction, aid in the design of liquid-supported microfluidic devices, and introduce new approaches to fabricating and modelling immersed LIS.

We disagree with the reviewer about moving these figures to SI. This is the first experimental evidence of nanobubbles on LIS, and Figures 4a and 4b constitute the basis to support our conclusions. It is not possible to achieve this information with optical techniques, therefore, AFM is the only technique to characterize nanobubbles on structured surfaces. This is the first time we show we can map nanobubbles on a lubricant-infused surface under water, a substantial extension of our previously developed AFM method. Given the importance of the AFM data, Figure 4a was included on recommendation of Prof. Hans-Jürgen Butt, who provided feedback on the paper prior to submission.

Reviewer #2:

Reviewer comment:

1. Please discuss in detail the advantages of using nanobubbles on LIS to achieve drag reduction compared with the traditional superhydrophobic surfaces. These contents will increase the impact of the article.

Answer: We have added a clarifying statement in the manuscript noting the advantages of LIS compared to superhydrophobic surfaces and added a reference to our published work on this point on page 9. LIS have been shown to be superior to superhydrophobic surfaces as marine antifouling surfaces, and this work shows that they show significant drag-reduction, comparable to superhydrophobic surfaces. We have modified the introduction to clarify that the aim of our work was to investigate in depth the mechanism of fluid slip on lubricant-infused surfaces. Therefore, we are not 'using' nanobubbles to enhance slip in LIS, we revealed that nanobubbles spontaneously nucleate in LIS and explain the experimental observations of large slip in LIS.

Reviewer comment:

2. In the experimental method, the author described how to control the concentration of gas in water. By placing water in a chamber that containing gas with a certain pressure, until it reached an equilibrium state. It is worth noting that it takes a long time for the gas molecules to diffuse into the water to reach equilibrium when the water is at rest. 30 mins may not be enough for them to reach equilibrium. According to the reviewer's experience, the concentration of the gas will only reach 90% of the equilibrium value overnight. So, if possible, consider using an oxygen meter to monitor the gas concentration in the solution. Authors can refer to methods in the reference.1

Answer: We measured the oxygen content in the water prepared according to our pressurising methods. We have modified Figure 1 to include horizontal error bars in the value of air content in each case, and modified the values of air content throughout the paper. The oxygen meter measurements revealed that the air content after at least 30 minutes of applied pressure is within 80 - 90% of the estimated value initially provided based on Henry's law. We have added a paragraph in methods and an extensive discussion of these measurements in SI (pages 2-4).

Reviewer comment:

3. In some Figures, the authors give the AFM images in the same position under different conditions. Please briefly introduce how to repositioning in different experiments.

Answer: In Fig. 4 and Fig. S10 (in new manuscript), when mapping the evolution of the gas nucleation process over time, we continuously mapped the same region with AFM tip engaged in water, without re-engaging or re-positioning the tip until the nanobubble appeared. Therefore, repositioning was not necessary.

Reviewer comment:

4. Please explain why the drag reduction is still obvious when using degassed water, as shown in Figure 3. Because nanobubbles should dissolve in degassed water. Does the TW-air surface have drag reduction when degassed water is used?

Answer: The conditions used for degassing do not completely remove gas from water, as quantified using the oxygen sensor. Secondly, bubbles could be formed when first filling the microfluidic channel with water, which is consistent with our observation that the pressure drop reduction remaining constant throughout our measurements. This effect has been observed before. For example, Watanabe et al. (J. Fluid Mech, 381, 1999) found that the drag reduction of superhydrophobic surfaces did not change when using saturated and degassed water.

Reviewer comment:

5. Although it is generally believed that the bubble surface cannot sustain shear stress, some experiments show that this boundary condition is very sensitive to the adsorption on the gas-liquid interface. When a small amount of surfactant (or contaminants) is adsorbed on the gas-liquid interface, the gas-liquid interface may become no-slip². Please briefly comment on the influence of this effect on the experimental results.

Answer: As discussed in reply to point 5 by reviewer #1, it is expected that our nanobubbles are cloaked by a thin layer of silicone oil because the spreading parameter S_{ow} of the oil over the water in air is positive: $S_{ow} = \gamma_{wv} - \gamma_{ov} - \gamma_{ow} = 10.5 \text{ mN/m}$, where $\gamma_{wv}, \gamma_{ov}, \gamma_{ow}$ are the interfacial energies of the water-vapor, oil-vapor and oil-water interfaces, respectively. We are aware on the literature on the effect of surfactants creating a Marangoni flow at the interface of bubbles/droplets and eliminating slip. The effect of an oil layer such as the one present here is not expected to be identical to that of surface-active molecular layer, and the drag reduction results certainly indicate that the slip is not eliminated by the presence of silicone oil. There is plenty of evidence from different research groups of drag reduction by air-infused superhydrophobic surfaces similar to ours, including with nanostructured surfaces (e.g., see Joseph et al. (2006) Phys. Rev. Lett., 97(15), 1–4), even in the potential presence of traces of contamination.

Reviewer #3:

Reviewer comment:

I am wondering that they did not find evident slip change on OTS coated silicon wafer. It is suggested that authors should measure the results of slip on OTS wafer with a large content of air in water.

Answer: This would be an interesting experiment to do. Nanobubbles have been reported on smooth OTS-silicon surfaces (Zhang et al., Langmuir 22, 2006; Ishida et al. Langmuir 16,

2000; Ducker, Langmuir 25, 2009), however, very high gas content is not required for nanobubbles to appear (see Seddon *et al.*, PRL, 2011). Under the highest gas content tested in our experiments, it is possible that nanobubbles are present on the smooth substrates but are not sufficient to reduce hydrodynamic drag. This could be explained in two ways: 1) they are not stable under flow given the flat nature of the underlying Si wafer (as opposed to the rough Teflon wrinkles), or 2) nanobubbles on OTS have a large contact angle (as reported by Zhang *et al.*, Langmuir 22, 2006), therefore their geometrical configuration causes low to no slip as reported by Steinberger *et al.* for protruding gas mattresses (Nature Materials, 6, 2007).

Carrying out experiments with highly gassed water in our microfluidic setup is problematic due to nucleation of macroscopic gas bubbles on the walls of the microfluidic channel and tubing, making the pressure drop measurements highly variable.

Reviewer comment:

Dissolved oxygen concentration may more precise than estimated by Henry's law.

Answer:

As discussed in reply to point 2 by reviewer #2, we have measured the oxygen content and confirmed that our estimated based on Henry's law were mostly correct. We have amended Figure 1 and the text to include the new air content values and error bars.

Reviewer comment:

Please check the Figure 3c in page 5, "blue circle in Fig.3c" or Fig.3e? Also, ...as a dashed line in Fig.1 and Fig.3c..., here Fig.3c or Fig.3e?

Answer: These typographical errors were corrected.

REVIEWER COMMENTS

Reviewer #1 (Remarks to the Author):

No further comments.

Reviewer #3 (Remarks to the Author):

No further comments.